# Evaluation of Groundwater Flow Changes Associated with Drainage within Multilayer Aquifers in a Semiarid Area

**Li Chen** [1,2]**, Xiaojun Wang** [1,2,]*****, Gelong Liang** [3] **and Haicheng Zhang** [3]

[1] State Key Laboratory of Hydrology-Water Resources and Hydraulic Engineering, Nanjing Hydraulic Research Institute, Nanjing 210029, China
[2] Research Center for Climate Change, Ministry of Water Resources, Nanjing 210029, China
[3] Hetaoyu Coal Mine, Huaneng Qingyang Coal and Power Co., Ltd., Qingyang 745000, China
***** Correspondence: xjwang@nhri.cn

**Abstract:** In order to evaluate the impact of groundwater drainage on groundwater flow, the Hetaoyu coal field was taken as a case study in the Longdong area, China, where the coal seam was covered with multilayer aquifers. A three-dimensional unsteady groundwater flow model and a one-dimensional fracture water flow model were calculated by joint equations for changing hydrogeological structures under coal mining. According to the results, mine construction had greatly affected groundwater reserves in the Quaternary phreatic aquifer, Cretaceous Huanhe confined aquifer, and Luohe confined aquifer. The groundwater drainage was mainly from the Cretaceous aquifer, in which the aquifer reserves of the Luohe Formation decreased by 30,861.8 $m^3$/m, accounting for about 92% of the total changes in local groundwater reserves. A drop funnel with an area of about 2.3 $km^2$ would be formed under the groundwater discharge of 187.6 $m^3$/h for the main inclined shaft excavation of the Hetaoyu coal mine. With the continuation of mining activities, the mine water flow will reach 806.83 $m^3$/h and would result in descending funnel area of about 4.5 $km^2$, the groundwater level drawdown at least 16 m, which would exceed the limited value regulated by the government. Therefore, in order to ensure the safety of coal mining and protect groundwater resources, the Hetaoyu Coal Mine departments should take some water loss prevention and control projects to reduce the drawdown of groundwater.

**Keywords:** groundwater flow; groundwater drainage; numerical model; water-conducted fracture zone; semiarid area

## 1. Introduction

The groundwater resource is critical for human survival and economic development in the semiarid area of Northwest China, where the coal seam and oil-bearing formation are usually overlaid by multilayer aquifers [1]. However, for coal and oil safety production, a large amount of groundwater should be discharged to the surface [2], which would influence local groundwater storage [3]. In order to protect the groundwater resource, the groundwater flow was usually simulated before coal mining [4]. Owing to the complexity of hydrogeological conditions in the mining area, the hydrogeological parameters vary strongly with time and space [5], and even the geological structure could change with mining [6]. So, it was difficult to estimate groundwater flow for each aquifer accurately.

There were several methods to estimate groundwater flow, such as the numerical model, analytical method, water balance method, and hydrochemistry method [7].

The numerical model is based on observed geological data [8] and can adjust the parameters conveniently when hydrogeological conditions change. Because of the groundwater drainage, the balance was disturbed, which resulted in the hydraulic connection changes between previously separate aquifers [9]. Based on the traditional groundwater numerical model, some scholars attempted to adjust parameters to solve the

discontinuity in the groundwater model when the new water-connected zone formed. Sun et al. (1989) first introduced two correction factors to the hydrology conductivity model when groundwater went through a fault [10]. Liang et al. (2012) encrypted the numerical model grids for the fault and then assigned different values to each node [11]. Based on the previous study, Karimzade et al. (2017) established the probability distribution functions for the fracture size, fracture orientation, and fracture density, respectively. According to these functions, flow paths in the rock were identified [12]. Mu et al. (2020) established a numerical couple model between fluid and solid matter to simulate the fault-zone deformation and water inflow under natural and disturbed conditions [13]. Song and Liang (2021) applied the numerical simulation method to analyze the process of fracture activation driven by mining, and the results showed good agreement between the theoretical calculation and the numerical simulation [14]. Xue et al. (2022) established a coupling model of gas drainage and groundwater loss to analyze the groundwater loss [15]. Liu et al. (2020) used the mixing calculations to identify sources of groundwater recharge in a Coastal Gold Mine [16]. Therefore, the numerical model has been widely used in groundwater inflow simulation, but there is much less research on coupling the 3D groundwater flow and 1D fracture water flow in a mining area.

Compared with the numerical model, the water balance method is based on the principle of mass conservation, but it requires more hypotheses for groundwater prediction [17]. Based on groundwater dynamics in a certain equilibrium period, the relationship between charge and discharge items was established, so the equilibrium equation can be used to predict the mine water inflow, which is suitable for the water-filled deposit with nonseepage groundwater inflow and simple water equilibrium conditions [18]. Elmarami et al. (2017) pointed out that the water balance method was suitable for large-scale calculation. The smaller the scale, the greater the uncertainty of each equilibrium term [19]. The groundwater flow calculated by this method can only be used as a reference value, not as an accurate result. In addition, climate change would have a significant effect on groundwater recharge [20]. If the recharge amount is smaller than other equilibrium terms, it will be very greatly influenced by evaporation. So, the water balance method is not suitable for fine calculation in the mining area with a multilayer aquifer.

On the other hand, the hydrochemistry method was useful for identifying the source of groundwater drainage by hydrochemical tracer test [21]. Qiao et al. (2019) took 76 groundwater samples for hydrogeochemical analysis, which indicated that coal mining caused the rock strata to sink, obstructing the water connection between aquifers [22]. Duan et al. (2019) identified the source of the coastal mine water inrush using hydrogeochemical analysis and inferred the potential water flow channels and the prevailing supply patterns [23]. Pilla et al. (2006) adopted hydrochemistry and isotope geochemistry as tools to define the recharge mechanisms, the origin of groundwater, and the hydraulic confinement of deep aquifers [24]. Powell et al. (2003) assessed the microbial quality of groundwater within the Permo-Triassic Sherwood Sandstone aquifers underlying the cities of Nottingham and Birmingham, and the groundwater sources were detected by microbial quality [25]. However, the hydrochemical method could not reveal the mechanism of groundwater flow, and it is difficult to couple the model with groundwater flow and fracture water flow in changing environment.

Therefore, the numerical model is closely associated with hydrogeological conditions, which could simulate the groundwater flow more accurately. So, in this paper, the coupled model of three-dimensional unsteady groundwater flow and one-dimensional fracture water flow was established based on hydrogeological borehole data, groundwater level, and so on. The hydrogeological structure model would be modified when the new water-conducted zone formed. The results would provide some guidance on the utilization and protection of local groundwater resources.

## 2. Materials and Methods

### 2.1. Study Area

The Hetaoyu coal field is located in Zhengning county in the southeastern city of Qiangyang, Gansu Province. It belongs to the Longdong Loess Plateau, and the landform is mainly composed of the Loess plateau area, Loess ridge, and valley area. As shown in Figure 1, the altitude is generally 889 to 1310 m. The plateau height decreases gradually from 1300 m to 1170 m with higher elevation in the north and east. The valley area mainly consists of Jinghe River and its tributaries, including Wuritiangou River and Silang River. Jinghe River Valley is about 210 m lower than the Loess Plateau.

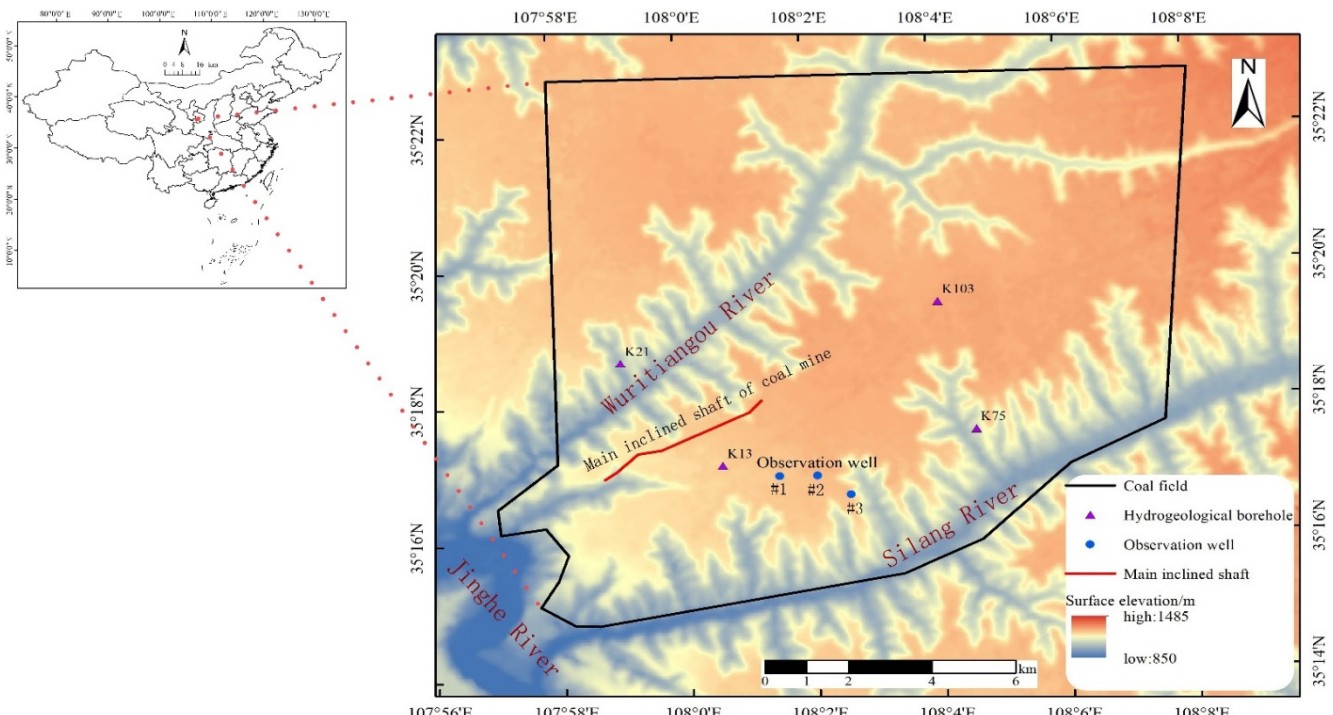

**Figure 1.** The location of the Hetaoyu coal field.

It belongs to a semihumid and semiarid climate, with an annual average precipitation of 598.44 mm, annual average evaporation of 1496.66 mm, and average relative humidity of 56–75%. Precipitation is mainly concentrated from July to September. The frost-free period is 101–117 days, and the maximum permafrost depth is 89 cm.

There are three exploitable coal seams in this coalfield, named coal-2 layer, coal-5 layer, and coal-8 layer. The design production capacity of the coal mine is 12.0 Mt/a. The main inclined shaft for coal mining is along the southwest direction, and the vertical angle is 7°. The Quaternary aquifer, Huanhe Confined aquifer, and Luohe Confined aquifer would be exposed in sequence, and then a large amount of the groundwater would be discharged to the surface. So, the groundwater around the main inclined shaft would be affected seriously, but further analysis of its impact is necessary for evaluating groundwater flow changes.

Therefore, in order to analyze the effect of the main inclined shaft excavation on the groundwater and the groundwater dynamics under coal-8 mining in the future, the study area took the main inclined shaft as the center; other boundaries were parallel or vertical to the equal-head line of the confined aquifer of Luohe Formation, which is situated at 107°59′2″–108°04′32″ E and 35°15′59″–35°20′3″ N, with an area of 32.3 km² within the Hetaoyu coalfield. There were four geological boreholes and three hydrological holes in this area, named #1, #2, and #3, as shown in Figure 2. The #1 recorded the groundwater level in Luohe Confined aquifer from November 2019 to May 2021. The #3 recorded the

groundwater level in Huanhe Confined aquifer from January 2019 to November 2019. The #2 recorded the mixed groundwater level of the Quaternary aquifer, Huanhe Confined aquifer, and Luohe Confined aquifer from November 2019 to May 2021. The isolevel map of the Luohe Formation aquifer is shown in Figure 2.

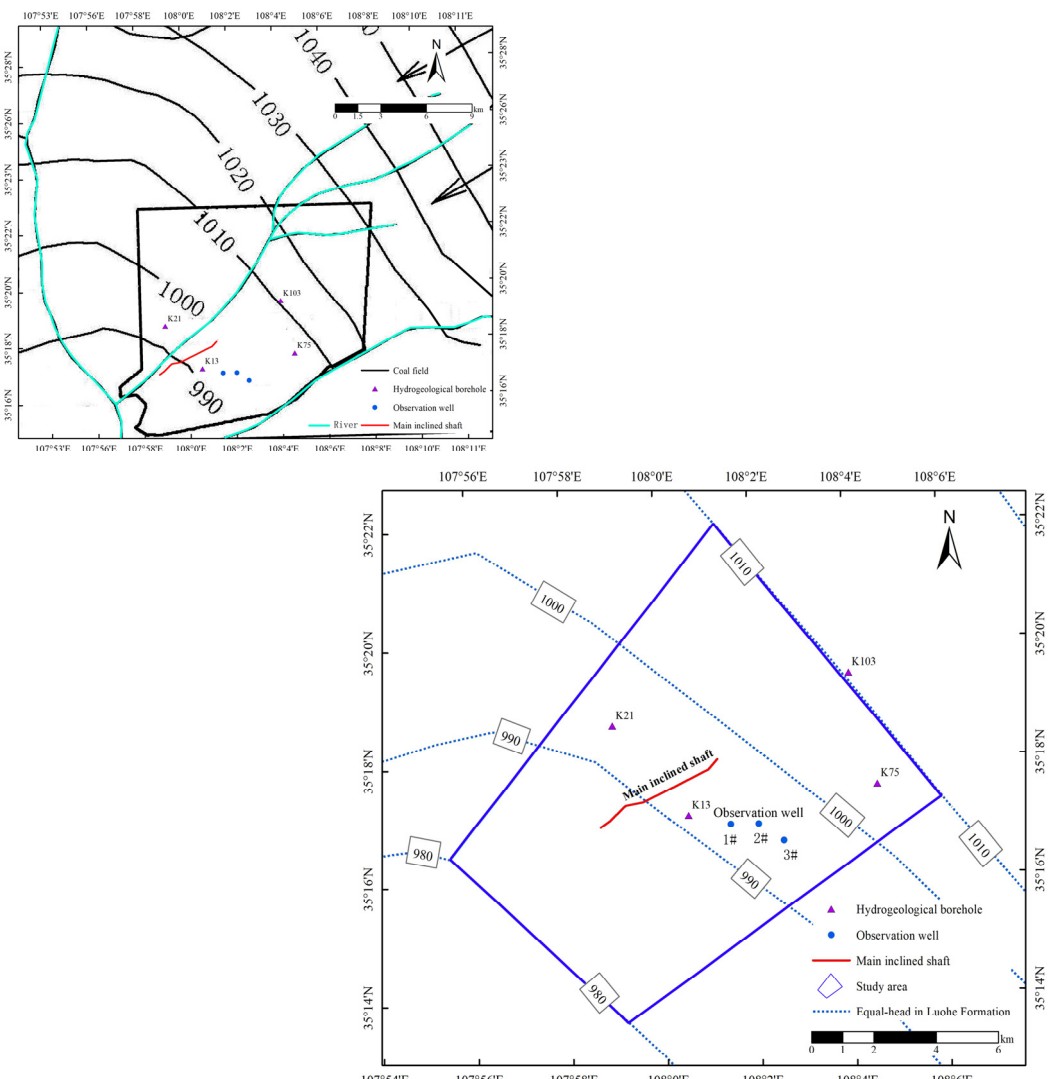

**Figure 2.** The schematic diagram of the study area and the isolevel map of the Luohe Formation aquifer.

The vertical geological structure was divided into seven layers, including three aquifers, three aquicludes, and a coal seam, by integrating similar and thin lithologies. The aquifers are Quaternary phreatic aquifers, Cretaceous Huanhe Formation confined aquifers, and Cretaceous Luohe Formation confined aquifers. The hydrogeological section is shown in Figure 3. Because of the discontinuous distribution of aquifers in the Huanhe Formation and Luohe Formation, there is a need to consolidate the aquifers within the same layer for convenient calculation. The Quaternary phreatic aquifers are composed of loess, and confined aquifers in the Cretaceous are composed of siltstone, fine sandstone, coarse sandstone, medium sandstone, and medium conglomerate. The aquicludes are mainly silty mudstone, mudstone, and clay.

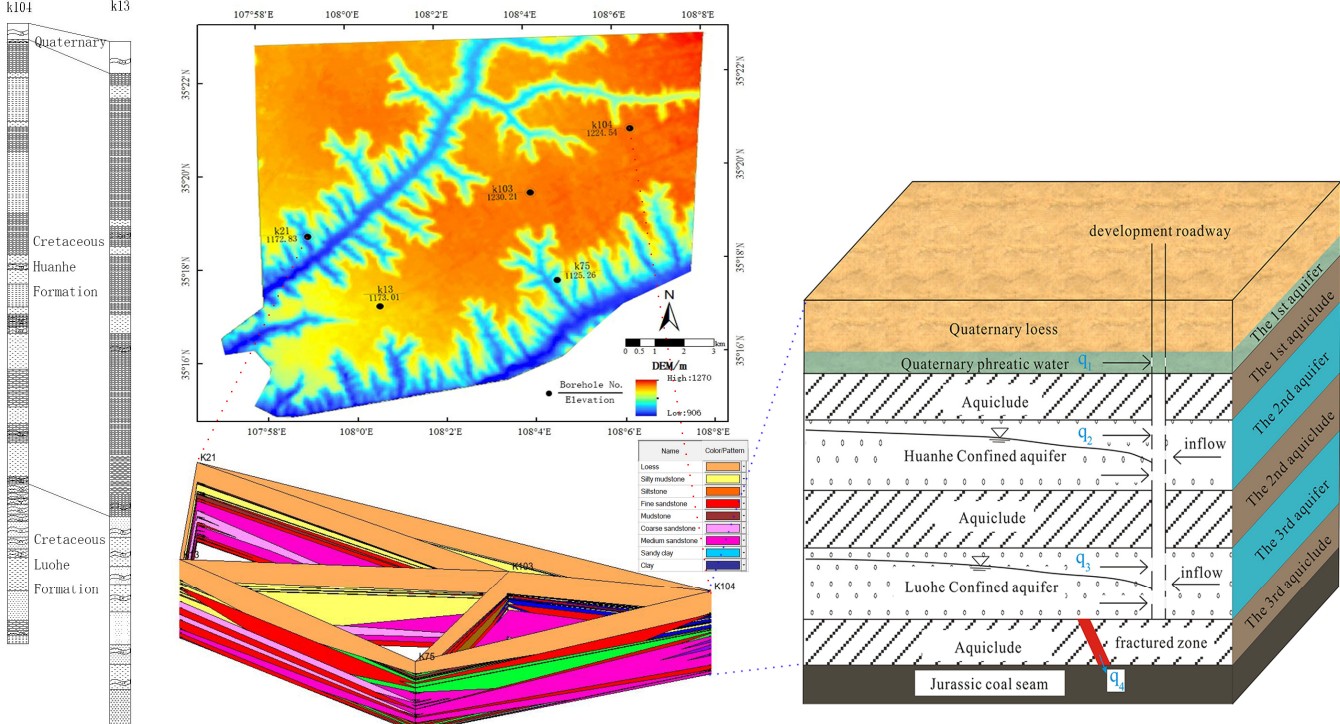

**Figure 3.** Diagram of the hydrogeological section of the mining area.

## 2.2. Model Description

There are several methods to estimate groundwater dynamics, such as the numerical model, analytical method, water balance method, mathematical, statistical method, and so on. However, the numerical model can not only describe the aquifer system and boundary conditions more realistically but also couple with different models more flexibly by writing computational codes. So, a three-dimensional groundwater dynamics model was adopted in this paper. Using a Cartesian coordinate system, the general equation for water flow with saturated, heterogeneous, and anisotropic porous media is expressed as follows

$$\frac{\partial}{\partial x}\left(K_x \frac{\partial H}{\partial x}\right) + \frac{\partial}{\partial y}\left(K_y \frac{\partial H}{\partial y}\right) + \frac{\partial}{\partial z}\left(K_z \frac{\partial H}{\partial z}\right) + W = S_s \frac{\partial H}{\partial t} \tag{1}$$

where $H$ is the total head, $W$ is the recharge rate, $S_s$ is the specific storage of the porous medium, $K_x$, $K_y$, $K_z$ are the saturated hydraulic conductivity in $x$, $y$, and $z$ directions, respectively.

According to the pumping test results from the Huanhe Formation confined aquifers and Luohe Formation confined aquifers, the initial values of hydrological parameters are listed in Table 1.

**Table 1.** Hydrogeological parameters of the main strata lithology in the mining area.

| Lithology | Saturated Hydraulic Conductivity m/d | Porosity | Specific Yield | Storage Coefficient m⁻¹ |
|---|---|---|---|---|
| Loess | 0.036 | 0.49 | 0.18 | 0.02 |
| Silty mudstone | 0.001728 | 0.21 | 0.12 | 0.000003 |
| Siltstone | 0.00000864 | 0.05 | 0.21 | 0.000003 |
| Fine sandstone | 0.0001728 | 0.1 | 0.23 | 0.000003 |
| Mudstone | 0.001728 | 0.21 | 0.12 | 0.000003 |
| Coarse sandstone | 0.0008728 | 0.3 | 0.27 | 0.000003 |
| Medium sandstone | 0.0015728 | 0.2 | 0.25 | 0.000003 |
| Medium conglomerate | 0 | 0.1 | 0.13 | 0.000003 |
| Clay | 0.000864 | 0.45 | 0.03 | 0.001 |

A one-dimensional fracture water flow model under the dominant water-conducted direction is usually used to represent the flow within the water-conducted zone. The numerical model is as follows, based on Navier–Stokes Equation

$$q_f = \frac{\gamma}{4\mu}J_f \int_0^{b/2}(b^2 - 4y^2)dy = \frac{g\delta^3}{12\mu}J_f \tag{2}$$

$$v_f = \frac{g\delta^2}{12\mu}J_f = K_f J_f \tag{3}$$

where $q_f$ is the unit discharge within the water-conducted zone, $v_f$ is the average flow velocity within the water-conducted zone, $g$ is the groundwater specific gravity, is the width of fracture, $\mu$ is the kinetic viscosity coefficient of water, $K_f$ is the conductivity of fracture, $J_f$ is the hydraulic gradient of flow.

The permeability changes are associated with mining-induced stress and water pressure. It is usually a negative relationship between hydraulic conductivity and the principal stress of fracture zones described by a negative exponential function [26]. It is a positive relationship between hydraulic conductivity and pore water pressure of fracture zones. The relationship could be expressed as follow [13]

$$K_f = K_0 e^{-\alpha(\sigma_m - p)} \tag{4}$$

where $\sigma_m$ is the mean principal stress, $\alpha$ is a fitting parameter, $p$ is the pore water pressure, and $K_0$ is the hydraulic conductivity.

The maximum height of the water-conducted fracture zone can be calculated by Equation (5)

$$h_{cond} = m - \sum_{n=1}^{k-1} h_n(\gamma_n - 1) \tag{5}$$

where $h_{cond}$ is the maximum height of water conducted zone, $m$ is the thickness of the coal seam, $n$ is the number of rock layers, $k$ is the total number of rock layers, $h_n$ is rock thickness at n layer, and $\gamma_n$ is the coefficient of rock-breaking expansion.

Equation (1) can be discretized by Equation (6) with a finite difference scheme for isotropic porous media. The calculation nodes are shown in Figure 4.

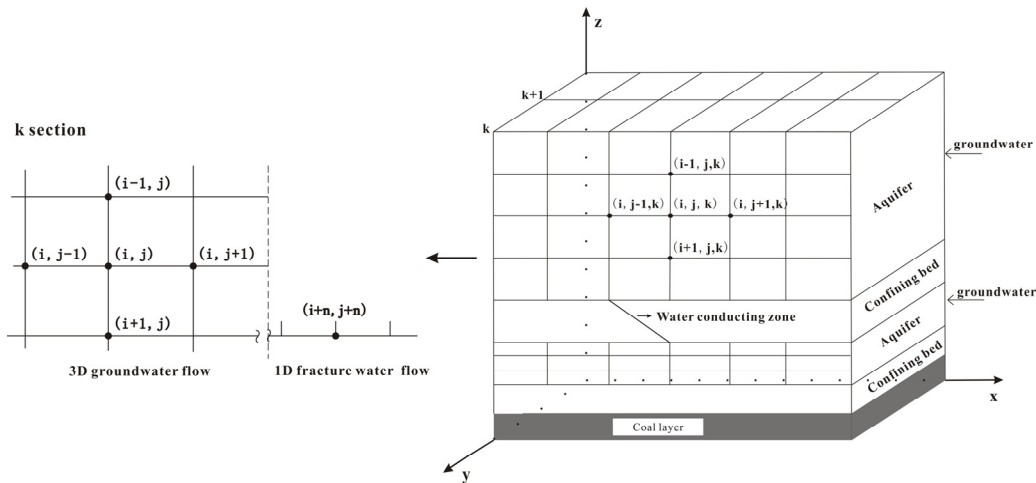

**Figure 4.** Schematic diagram of calculation nodes of the groundwater flow model.

$$\frac{K_x}{\Delta x^2}\left(H_{i,j-1,k}^n - 2H_{i,j,k}^n + H_{i,j+1,k}^n\right) + \frac{K_y}{\Delta y^2}\left(H_{i,j,k-1}^n - 2H_{i,j,k}^n + H_{i,j,k+1}^n\right)$$
$$+ \frac{K_z}{\Delta z^2}\left(H_{i-1,j,k}^n - 2H_{i,j,k}^n + H_{i+1,j,k}^n\right) + \sum W^n = \frac{S_s}{\Delta t}\left(H_{i,j,k}^n - H_{i,j,k}^{n-1}\right) \tag{6}$$

Where $K_x$, $K_y$, $K_z$, and $S_s$ are constant in each aquifer, $n$ and n – 1 are the time at n and n – 1, $H_{i,j,k}^n$ is the total head of node , $i, j, k$ at $n$ time, and $\sum W^n$ is the total recharge rate at time n. In this paper, $W$ is the mine drainage varied with time during main inclined shaft excavation, as shown in Figure 5.

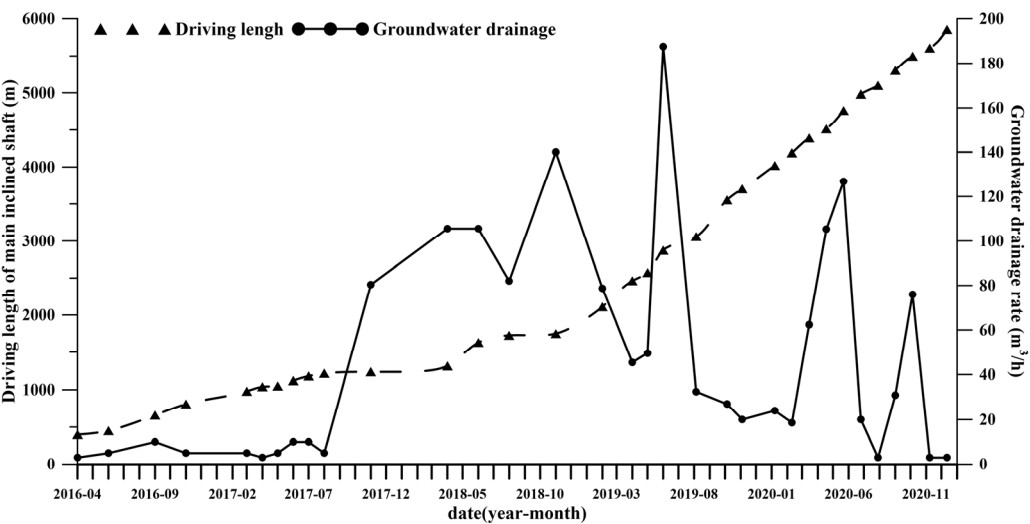

**Figure 5.** Time variations of groundwater drainage with coal mining in Hetaoyu Coal Mine.

## 3. Results and Discussion

### 3.1. Spatial and Temporal Variation of Groundwater Level for Different Aquifers

The variations of groundwater levels for different aquifers are shown in Figure 6a,b. It could be noticed that the trends of groundwater level varied from each other, especially in #3 the groundwater level had been rising with the increase in groundwater drainage. That phenomenon indicated that although there was a large amount of groundwater discharge in the coal mine area, the influence on regional groundwater flow was limited. The distance between the main inclined shaft and #3 was about 3 km, which might be the threshold of the drainage influence.

The Luohe aquifer in #1 presented a downward trend accompanied by groundwater drainage since November 2020, but the groundwater level was relatively steady with time

in #2 at about 1 km east of #1. That meant that the influence area of drainage might only reach near #1 and #2.

Therefore, although groundwater drainage caused the groundwater level to drop, the groundwater level at the location about 3 km away from the drainage area was not affected by the drainage. A similar phenomenon would occur in coastal areas, where the groundwater replenishes at a rapid rate. The groundwater level dropped by 29 m to 32 m after 7 days of pumping. After stopping pumping, the groundwater level would recover quickly, so the change in water level in a short time would not lead to large land subsidence and have little impact on the surrounding environment [27].

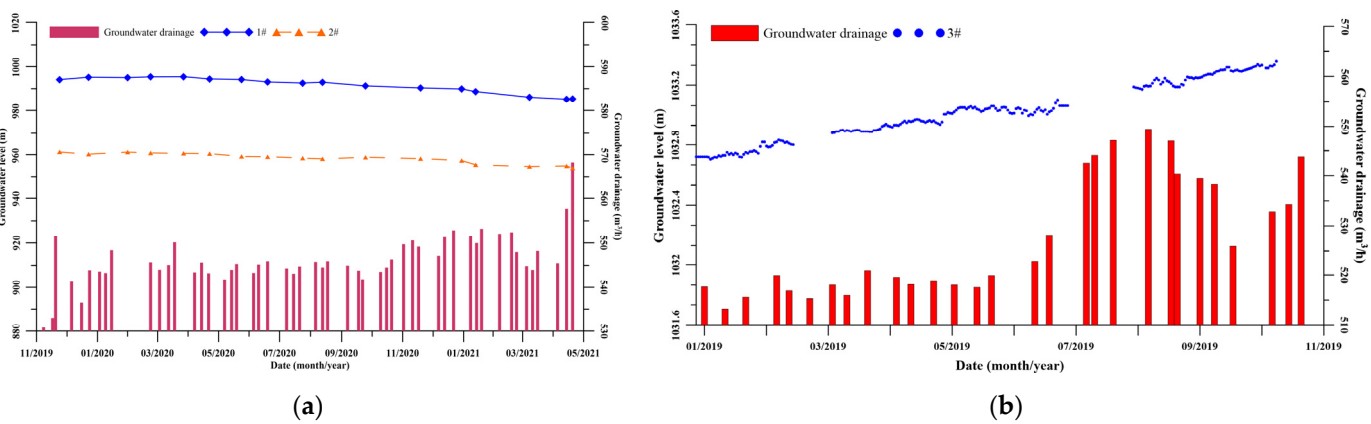

(**a**)　　　　　　　　　　　　　　　　　　　　　　　(**b**)

**Figure 6.** Temporal variations of groundwater level under groundwater drainage in Hetaoyu Coal Mine. (**a**) Luohe aquifer (1#) and mixed groundwater level (2#). (**b**) Huanhe aquifer (3#).

*3.2. The Relationship between Groundwater Levels of Various Aquifers*

It is well known that the impact of groundwater drainage is greatest in the mining area and gradually decreases with increasing distance. According to the above analysis, the variation of groundwater level in #2 was lagging behind that of #1. In order to quantify the lag time, the correlation coefficient was calculated between #1 and #2. The result is shown in Figure 7.

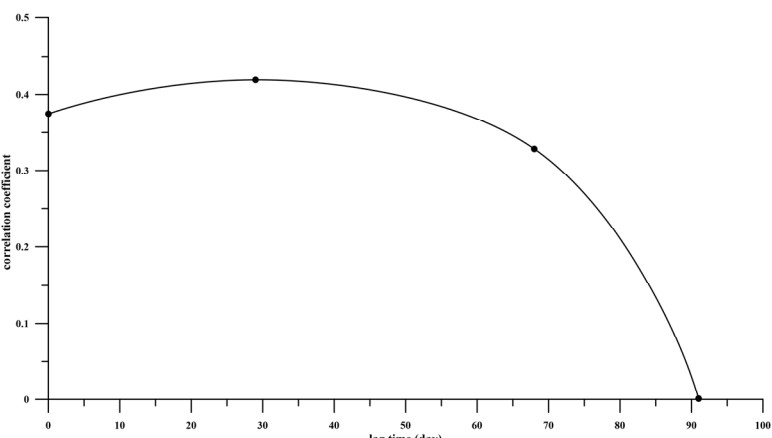

**Figure 7.** The correlation under different lag-time for #1 and #1.

As shown in Figure 7, the lag time maximum correlation coefficient appeared at 28d, indicating that under the groundwater drainage, each aquifer did not present a synchronous downward trend; after the first aquifer dropped, the near aquifer began to decline about 28d later. This meant that the groundwater flow was exchanged frequently in this area, and the losses of water would be recharged in a short time. This result also indicated

that the groundwater drainage in the mining area had little influence on the regional groundwater flow.

### 3.3. Influence on Local Groundwater Dynamics under Groundwater Drainage

Based on the above results, taking the groundwater drainage as a discharge item in Equation (6), the groundwater dynamic was analyzed, and the source of groundwater drainage was identified. The groundwater discharge of 187.6 m³/h was used as an example in Figure 5. The driving length of the main inclined shaft was drilled to the aquifer of the Luohe Formation in June 2019, and the impact of drainage on groundwater flow is shown in Figure 8.

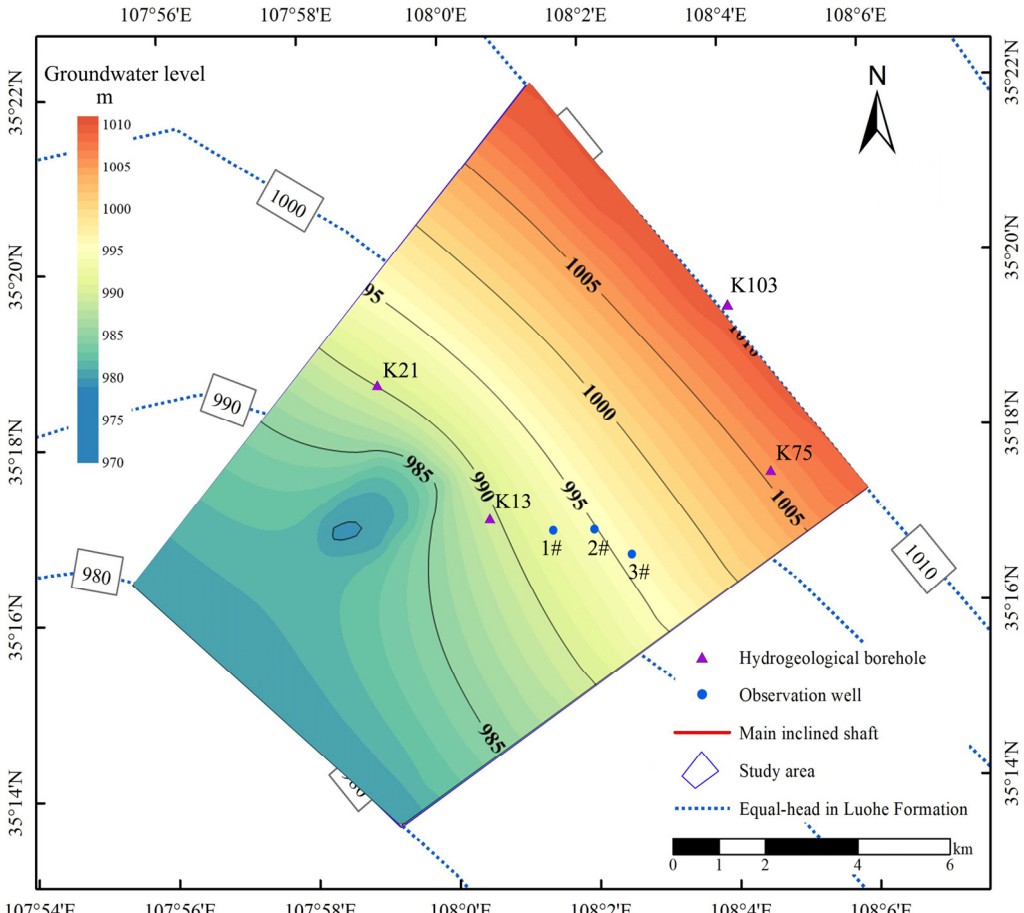

**Figure 8.** The water table contour map of the Cretaceous Luohe Formation aquifers under the groundwater discharge was 187.6 m³/h.

From Figure 8, with a large amount of groundwater discharging to the surface, the groundwater level of Luohe Formation dropped by about 14 m, and a descending funnel of about 2.3 km² was formed in the drainage area, which affected the characteristics of local groundwater flow. However, there was no change in the direction of groundwater flow on the regional scale, where the regional groundwater flow was discharged from the northeastern mountainous area to the Malian River. Nevertheless, a large amount of groundwater discharged into the Malian River was reduced as a result of groundwater drainage.

According to the drop depth of water level, the reserve changes of each aquifer are shown in Figure 9.

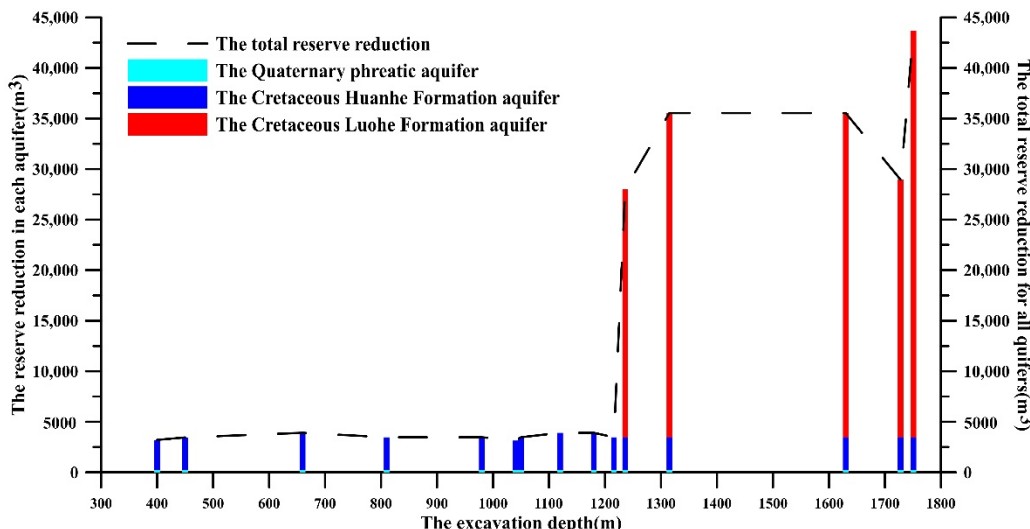

**Figure 9.** The time variation of aquifer reserve reduction under coal mining.

It showed that the Quaternary phreatic aquifer reserves and the Cretaceous Huanhe Formation aquifer reserves decreased by 249.3 m³/month and 3270.4 m³/month, respectively. When the driving length of the main inclined shaft was excavated into the Luohe confined aquifer, the groundwater drainage increased sharply, and the Cretaceous Luohe Formation aquifer reserves decreased by 30,861.8 m³/month, which accounted for about 92% of the total reserve reduction. Therefore, the Luohe Formation aquifer was the main source of groundwater drainage. In the process of coal mining, it is necessary to pay more attention to the influence of water flow from the confined aquifer of Luohe Formation and take corresponding water control projects. By plugging the groundwater recharge channel on fracture, the water yield in a coal mine could be reduced by 600 m³/h in a typical North China coal mine [28].

When the groundwater discharge was up to 250 m³/h, the water level of the Huanhe Formation aquifer and Luohe Formation aquifer dropped by about 11 m and 16 m, respectively. A descending funnel of about 2.4 km² was formed in the drainage area, which affected the characteristics of local groundwater flow. Based on the numerical calculation, the descending trend of groundwater level for Cretaceous Huanhe Formation and Luohe Formation aquifers under different groundwater discharges is shown in Figure 10.

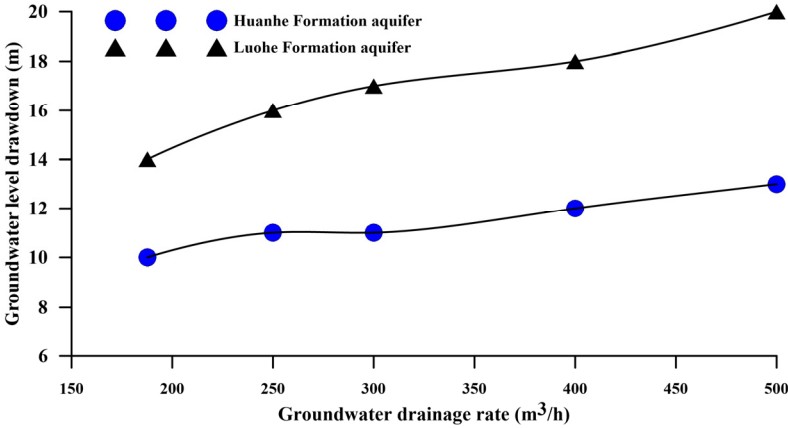

**Figure 10.** Groundwater level drawdown under the different groundwater drainage.

Compared with the groundwater level of the Luohe Formation in Tingnan Coal Mine near Hetaoyu Coal Mine, the groundwater level of the Luohe Formation dropped by 52.5 m from 2004 to 2015 [29]. However, with the improvement of water control works in each

coal mine recently, the groundwater level has declined more and more slowly. Zhang (2015) obtained similar results using the large well method [30].

Therefore, it can be inferred that the groundwater level would decrease with the increase in drainage, but the groundwater drawdown rate decreased gradually. The downstream groundwater level suffered more influence.

The results are of great significance to guide the construction of mine water-control projects, such as curtain grouting, to protect local groundwater resources. According to the groundwater level control requirements of the government, the mine water inflow could be adjusted by engineering technologies so that the groundwater level depth of each aquifer was in a controllable range.

*3.4. Prediction Groundwater Inflow Changes for Groundwater Drainage*

The average coal seam thickness of the coal-2 layer, coal-5 layer, and coal-8 layer was 1.87 m, 0.95 m, and 8 m, respectively. Based on geological boreholes and survey data, the results of the maximum height of the water-conducted fracture zone in the coal-2 layer, coal-5 layer, and coal-8 layer were calculated by Formula (5). The results are shown in Table 2.

**Table 2.** The maximum height of the water-conducted fracture zone of each coal seam in Haoheyu Coal Mine.

| Coal Seam | Distance from Quaternary Bottom Plate (m) | Normal Distance from the Bottom of the Lower Cretaceous Conglomerate (m) | Dip Angle of Coal Seam (°) | Coal Seam Thickness (m) | Height of Water Conducted Fracture Zone (m) |
|---|---|---|---|---|---|
| coal-2 layer | 690.93 | 79.04 | 6 | 5.05 | 64.34 |
| | 783.54 | 104.74 | 5 | 2.7 | 50.12 |
| | 764.85 | 88.79 | 6 | 2.21 | 28.35 |
| coal-5 layer | 734.98 | 90.18 | 6 | 8.86 | 86.78 |
| | 840.86 | 134.74 | 9 | 5.28 | 82.13 |
| | 799.47 | 123.41 | 6 | 2.4 | 35.17 |
| coal-8 layer | 748.49 | 121.89 | 7 | 23.56 | 186.94 |
| | 657.34 | 104.43 | 6 | 23.47 | 185.47 |
| | 832.02 | 155.96 | 6 | 13 | 182.63 |

As the driving length of the main inclined shaft increases, the new water conducting would appear until the height of the water-conducted fracture zone was larger than the thickness of the above aquicludes. Then, the above geological structure model and the hydraulic conductivity would change accordingly. According to the trends of the driving length in Figure 5, the future groundwater drainage was predicted by Equation (6). The simulation result is shown in Figure 11.

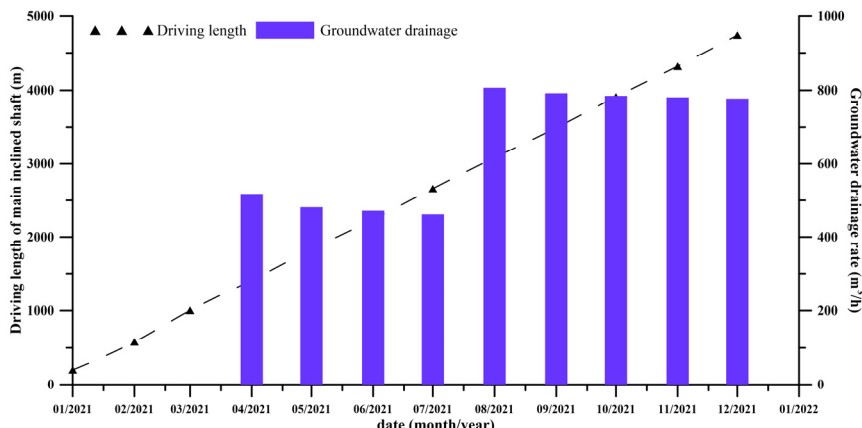

**Figure 11.** Prediction of the mine water inflow caused by the water-conducting fracture zone under coal mining.

With a linear increase in the driving length, when it reached 1410.4 m in the Hetaoyu coal field, the water-conducted fracture zone appeared, resulting in the groundwater drainage of 517.42 m³/h. It would generate another, new water-conducted fracture zone when the driving length reached 3083.2 m, and groundwater drainage increased to 806.83 m³/h. So, in order to ensure the safety of coal production, the drainage capacity of the pumps should be greater than 806.83 m³/h in future coal mining.

Taking the groundwater drainage with 806.83 m³/h as an example, the results are shown in Figure 12. The groundwater level of the Huanhe Formation aquifer and Luohe Formation aquifer would drop by about 16 m and 23 m, respectively. A descending funnel of about 4.5 km² would be formed in the drainage area, which would affect the characteristics of local groundwater flow. The groundwater discharging from the Luohe Formation aquifer would account for 86%, about 693.88 m³/h.

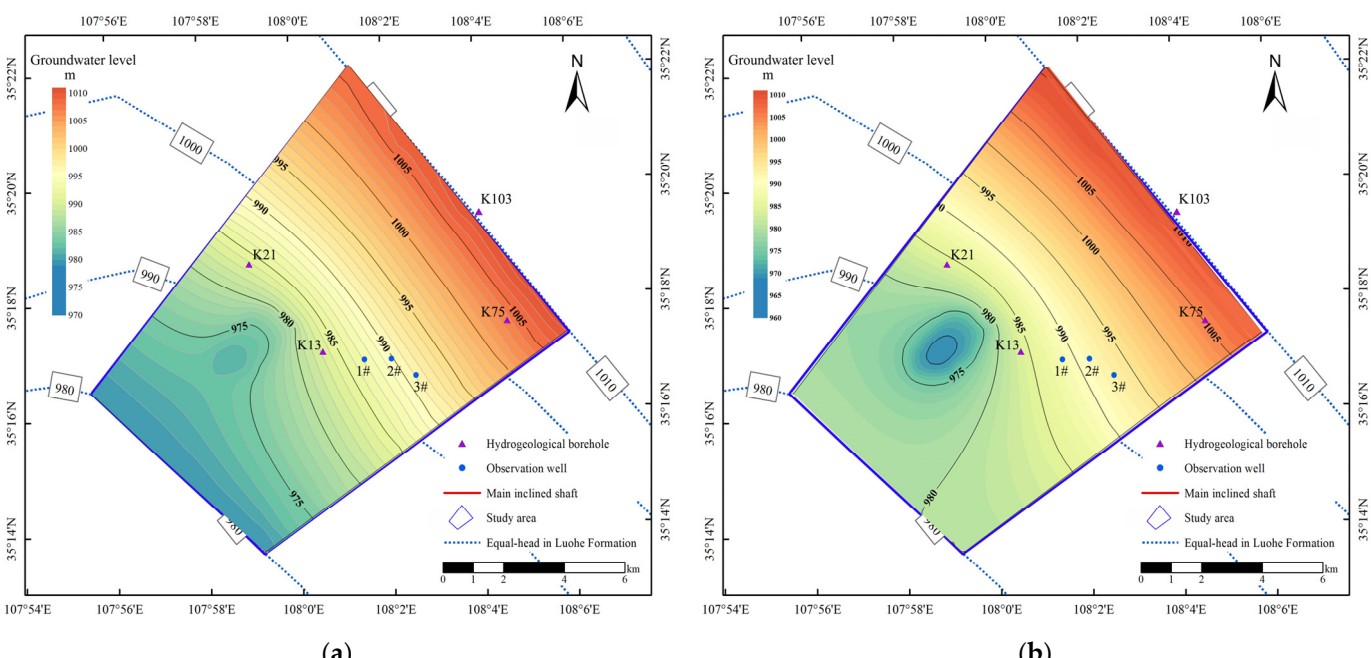

(**a**)                              (**b**)

**Figure 12.** The groundwater level contour map under the groundwater drainage rate was 806.83 m³/h: (**a**) the Huanhe Formation aquifers; (**b**) the Luohe Formation aquifer.

However, when the groundwater drainage rate increased to 806.83 m³/h, the regional groundwater level would not drop drastically. It meant that the local groundwater

drainage had little influence on the groundwater flow of the whole basin. Therefore, 4.5 km$^2$ around the drainage area was the key groundwater protection area, and the coal mine departments should strengthen the monitoring of groundwater level and fracture zone in this area.

*3.5. Uncertainty of Numerical Model*

The numerical model presents a good simulation of groundwater flow, but it usually simplifies the hydrogeological structure, parameters, and recharges for convenience calculation. So, there were some uncertainties in the coupled numerical model.

Firstly, as shown in Figure 3, the stratigraphic structure was divided into seven sections, but, in natural conditions, each lithology is discontinuous horizontally or vertically, and the aquifers are distributed in the form of multilayer nesting. However, in this paper, the aquifers were uniformly divided, and the nearby aquifers were consolidated. So, the geological structure had some uncertainties, but the total thickness of each aquifer in this paper did not change in order to reduce its influence.

Secondly, the strata are heterogeneous in the mining area, and the hydrogeological parameters vary with time and space. In this paper, the values of specific storage and hydraulic conductivity were derived from the pumping test at five holes, but there would be a scale effect on applying point data to the whole area. In order to avoid the influence of uncertainty of hydrogeological parameters, the parameters were verified by comparing them with the water table of the hydrological hole based on the observed values obtained by the pumping test. Moreover, according to the pumping test results at different layers, the parameter partitions were refined to ensure the reliability of the water inflow model.

In addition, the recharge item in Formula (1) is only the mine drainage. The rainfall infiltration, evapotranspiration, and irrigation pumping were not calculated; this paper focused on the impact of mine drainage on the groundwater system, thus simplifying other factors. Considering the thickness of the Loess layer of about 200 m in the mining area, rainfall infiltration and evapotranspiration had little influence on the aquifers. Based on the results of this paper, the irrigation pumping also had little influence on the Huanhe Formation aquifers and Luohe Formation aquifers. Generally, the uncertainty of recharge had a limited impact on the results.

Therefore, there were some uncertainties in the numerical model. Although the corresponding treatment was carried out, its influence on the results still needs further study.

**4. Conclusions**

The numerical model was used to estimate the groundwater dynamics with groundwater drainage in the Longdong area, where the groundwater was the main source for residents and agriculture. Based on the observation data and simulation results, the spatial and temporal variations in the groundwater level for different aquifers were analyzed, and the sources of groundwater drainage were quantified. Moreover, the groundwater flow under the influence of the water-conducted fracture zone was predicted for future coal mining.

(1) The groundwater drainage changed irregularly, which was not linearly related to coal production. It was mainly affected by the hydrogeological structure and lithological parameters. The groundwater level at the location about 3 km away from the drainage area was not affected by the drainage;

(2) When the driving length of the main inclined shaft was excavated in all aquifers, the Luohe Formation aquifer was the main source of the mine water inflow, which accounted for about 92% of the total water inflow. So, the Luohe Formation aquifer was the main object to be protected and monitored by the local government;

(3) During the coal mining period, when the new water-conducting formed, the mine water inflow would increase to 806.83 m$^3$/h under the influence of the water-conducted fracture zone, which would cause 23 m groundwater drawdown and descending funnel of about 4.5 km$^2$. This drawdown might exceed the limit value regulated by the Qingyang

Government. The Hetaoyu Coal mine departments should take some corresponding water prevention and control projects to reduce the drawdown of groundwater and pay more attention to monitoring the groundwater level within 4.5 km².

**Author Contributions:** Conceptualization, L.C., X.W., G.L. and H.Z.; methodology, L.C. and X.W.; formal analysis, L.C. and X.W.; writing—original draft preparation L.C.; writing—review and editing, L.C., X.W., G.L. and H.Z.; visualization, L.C. and X.W.; supervision, X.W. All authors have read and agreed to the published version of the manuscript.

**Funding:** This research was funded by the National Natural Science Foundation of China (Grant No. 51722905), Youth Fund Program of Basic Research Operating Expenses of Central Public Welfare Research Institute for Nanjing Hydraulic Research Institute (Grant No. Y520024), and Young Top-Notch Talent Support Program of National High-level Talents Special Support Plan and China Water Resource Conservation and Protection Project (Grant No. 126302001000150005).

**Institutional Review Board Statement:** Not applicable.

**Informed Consent Statement:** Not applicable.

**Data Availability Statement:** Data sharing is not applicable.

**Acknowledgments:** The authors are grateful to the students who participated in this study, Jiaqi Sun for collaborating on data collection and question development, and Wenke Wang for suggestions on an early version of this manuscript.

**Conflicts of Interest:** The authors declare no conflict of interest.

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
