# Peer review of "Evaluation of Groundwater Flow Changes Associated with Drainage within Multilayer Aquifers in a Semiarid Area"

_water, doi:10.3390/w14172679_

Round 1
Reviewer 1 Report
The authors offer a work that potentially can improve our knowledge about the topic of interest. I suggest that the authors consider a revision of their work along the following suggestions and questions.
Suggestions and Questions:
1-What are the advantages and disadvantages of this study? I suggest that the author emphasize this topic.
2-What are the limitations of this study? I suggest that the author emphasize this topic.
3-What is the effect of this study on practical projects? What are the benefits for future development? Authors are advised to elaborate.
4-The abstract of the article needs to be revised, and the importance of the article should be highlighted. There are insufficient references, so more references need to be supplemented. There are too few references, which need to be supplemented to 30. The background and mechanism of gas or water flow underground are not introduced clearly.
Y. Xue, J. Liu, X. Liang, S. Wang, and Z. Ma, “Ecological risk assessment of soil and water loss by thermal enhanced methane recovery: Numerical study using two-phase flow simulation,” Journal of Cleaner Production, vol. 334, article 130183, 2022.
Powell, K. L., Taylor, R. G., Cronin, A. A., Barrett, M. H., Pedley, S., Sellwood, J., ... & Lerner, D. N. (2003). Microbial contamination of two urban sandstone aquifers in the UK. Water Research, 37(2), 339-352.
Pilla, G., Sacchi, E., Zuppi, G., Braga, G., & Ciancetti, G. (2006). Hydrochemistry and isotope geochemistry as tools for groundwater hydrodynamic investigation in multilayer aquifers: a case study from Lomellina, Po plain, South-Western Lombardy, Italy. Hydrogeology Journal, 14(5), 795-808.
Reviewer 2 Report
1. Please explain the values of hydraulic parameters, e.g., Kx, Ky, Kz, Ss, Kf in your numerical model?
2. Please explain those aquifers more detail. Those aquifers are composed of sand, gravel, limestone? They are consolidated or unconsolidated or semi-consolidated? Horizontal or tilted?
3. Please discuss the uncertainty of your numerical model.
Round 2
Reviewer 1 Report
accept
Reviewer 2 Report
I have no further questions.